# The Impact of Lifestyle Interventions in High-Risk Early Breast Cancer Patients: A Modeling Approach from a Single Institution Experience

**DOI:** 10.3390/cancers13215539

**Published:** 2021-11-04

**Authors:** Mirco Pistelli, Valentina Natalucci, Laura Scortichini, Veronica Agostinelli, Edoardo Lenci, Sonia Crocetti, Filippo Merloni, Lucia Bastianelli, Marina Taus, Daniele Fumelli, Gloria Giulietti, Claudia Cola, Marianna Capecci, Roberta Serrani, Maria Gabriella Ceravolo, Maurizio Ricci, Albano Nicolai, Elena Barbieri, Giulia Nicolai, Zelmira Ballatore, Agnese Savini, Rossana Berardi

**Affiliations:** 1Department of Medical Oncology, Università Politecnica delle Marche, AOU Ospedali Riuniti di Ancona, 60126 Ancona, Italy; laura_sco@libero.it (L.S.); veroagostinelli@gmail.com (V.A.); edoardo_lenci@ospedaliriuniti.marche.it (E.L.); crocetti.sonia@alice.it (S.C.); merloni.filippo@gmail.com (F.M.); lucia.bastianelli@ospedaliriuniti.marche.it (L.B.); zelmira.ballatore@ospedaliriuniti.marche.it (Z.B.); agnese.savini@ospedaliriuniti.marche.it (A.S.); 2Department of Biomolecular Sciences, University of Urbino Carlo Bo, 61029 Urbino, Italy; valentina.natalucci@uniurb.it (V.N.); elena.barbieri@uniurb.it (E.B.); 3Dietology and Clinical Nutrition, AOU Ospedali Riuniti di Ancona, 60126 Ancona, Italy; marina.taus@ospedaliriuniti.marche.it (M.T.); daniele.fumelli@ospedaliriuniti.marche.it (D.F.); gloria.giulietti@ospedaliriuniti.marche.it (G.G.); claudia.cola@ospedaliriuniti.marche.it (C.C.); albano.nicolai@ospedaliriuniti.marche.it (A.N.); 4Department of Experimental and Clinical Medicine, Neurorehabilitation Clinic, Università Politecnica delle Marche, AOU Ospedali Riuniti di Ancona, 60126 Ancona, Italy; marianna.capecci@ospedaliriuniti.marche.it (M.C.); MariaGabriella.Ceravolo@ospedaliriuniti.marche.it (M.G.C.); 5Division of Rehabilitation Medicine, AOU Ospedali Riuniti di Ancona, 60126 Ancona, Italy; roberta.serrani@ospedaliriuniti.marche.it (R.S.); maurizio.ricci@ospedaliriuniti.marche.it (M.R.); 6Department of Medical Emergency, AOU Ospedali Riuniti Marche Nord, 61121 Pesaro, Italy; giulia.nicolai@ospedalimarchenord.it

**Keywords:** lifestyle, physical activity, Mediterranean diet, obesity, metabolic syndrome, anxiety, early breast cancer

## Abstract

**Simple Summary:**

High body mass index (BMI) is correlated with an increased production of hormones (estrogens, insulin, testosterone, leptin), and pro-inflammatory cytokines, which have been associated with breast cancer (BC) risk and recurrence. Regular physical activity (PA) decreases BMI and blood concentrations of testosterone, estrogens, insulin and pro-inflammatory cytokines. Moreover, the Mediterranean diet (MD) reduces obesity, metabolic syndrome (MS) and insulin resistance, which are all associated with increased risk of BC onset and recurrence. Despite the accumulating evidence of the detrimental effect of physical inactivity and overweight on BC recurrence, weight control and PA counseling are not yet current practice. The principal aim of the lifestyle intervention is to promote weight loss through diet and physical activity in overweight and/or high-risk breast cancer survivors. Moreover, implementing a “lifestyle interventions program” in clinical practice can lead to improvements in psychophysical well-being and favor the correction of cardiovascular risk factors and compliance with endocrine therapy, potentially translating into a prognostic advantage.

**Abstract:**

A healthy lifestyle plays a strategic role in the prevention of BC. The aim of our prospective study is to evaluate the effects of a lifestyle interventions program based on special exercise and nutrition education on weight, psycho-physical well-being, blood lipid and hormonal profile among BC patients who underwent primary surgery. From January 2014 to March 2017, a multidisciplinary group of oncologists, dieticians, physiatrists and an exercise specialist evaluated 98 adult BC female patients at baseline and at different time points. The patients had at least one of the following risk factors: BMI ≥ 25 kg/m^2^, high testosterone levels, high serum insulin levels or diagnosis of MS. Statistically significant differences are shown in terms of BMI variation with the lifestyle interventions program, as well as in waist circumference and blood glucose, insulin and testosterone levels. Moreover, a statistically significant difference was reported in variations of total Hospital Anxiety and Depression Scale (HADS) score, in the anxiety HADS score and improvement in joint pain. Our results suggested that promoting a healthy lifestyle in clinical practice reduces risk factors involved in BC recurrence and ensures psycho-physical well-being.

## 1. Introduction

Breast cancer (BC) is the most common tumor and the main cancer death cause among females worldwide. A woman’s life risk for BC has increased progressively from the 1930s to the 2000s. Currently, BC incidence is 24.5% and it is responsible for 15% of new cancer deaths [1]. Two risk factors groups are mentioned: the former group includes non-hereditary factors, and the latter group contains several inherited mutations, together with BC family and personal history. In fact, although the most common BCs are sporadic, 5–7% of cases are hereditary, and BRCA1 and BRCA2 genes mutations have a predominant role [2]. Among demographic characteristics, gender and age are the most relevant risk factors; moreover, early menarche, late menopause, delayed first pregnancy and low parity are important reproductive factors increasing BC risk [3]. Conversely, pregnancy and breast-feeding represent protective factors [4].


*“If we were able to provide everyone with the right amount of nutrition and exercise, neither in excess nor in fault, we would have found the way to health”. Hippocrates, 460–377ac.*


A healthy lifestyle plays a strategic role in the prevention of cardiovascular, lung and cancer diseases, including BC. In fact, according to the World Health Organization (WHO), avoiding unhealthy lifestyle (i.e., tobacco use, inactivity and obesity) in addition to intake of adequate fruits and vegetables, could prevent more than 30% of cancer deaths [5]. Alcohol and animal fats are linked to an increased risk of BC [6]. Obesity and metabolic syndrome (MS) seem to be important risk factors, too [6], while the Mediterranean diet (MD) plays a protective role in cancer onset [7,8]. Since its identification in 1986 by Keys et al., MD appeared as a healthy diet. Turati et al. conducted a multicentric hospital-based case-control study, collecting data from 3034 women with a BC diagnosis (case), admitted to several Italian and Swedish hospitals and without other cancer history, along with 3392 patients as controls. This latter group included patients admitted to the same areas with a wide spectrum of conditions, but without anamnestic cancer history. The authors administered a structural questionnaire including information on lifestyle and dietary habits. They highlighted that, as acknowledged for several other cancers, adhering to the Mediterranean dietary pattern correlates to a reduced risk of BC [7]. Other papers strengthened this idea. In Schwingshackl’s et al. systematic review, the authors demonstrated a protective role of MD, able to lower cancer risk and mortality in particular in breast, colorectal, liver, gastric, head and neck, gallbladder and biliary tract cancer patients, commonly due to fruit and vegetable fasting [8].

Avoidance of excessive fat intake is the most important protective factor to prevent body fat accumulation [9]: obesity correlates with a 35% to 40% increased risk of BC recurrence and death, especially in estrogen receptor-positive (ER+) BC [10]. It is consistently underlined that a fat-rich diet stimulated cancer growth: in overweight patients, BC is most often diagnosed in advanced stages or with high-grade histology and the adjuvant endocrine therapy or chemotherapy seem to have less efficacy. Obesity is also an independent prognostic factor for developing metastases, also after ten years [11]. Several preclinical and clinical studies confirmed that preventing the accumulation of body fat by calorie restriction (CR, reduction of dietary energy intake by approximately 30% without malnutrition) acts as a protective factor for chronic conditions such as cardiovascular disease, diabetes and cancer [9]. In particular, the insulin-like growth factor-1 (IGF-1) signaling pathway has a noteworthy role in cancerogenesis. Biologically, IGF-1 is a nutrient-responsive growth factor triggering Ras/MAPK and PI3K/AKT kinases. The activation of the first pathway promotes nuclear changes, finally responsible for transcription of genes involved in cellular growth and proliferation, while activation of PI3K/AKT pathways acts by decreasing apoptosis and by promoting glucose metabolism. Since cancer cells use IGF-1 pathway as a stimulus for proliferation and growth, its reduction in patients practicing CR translates into a decreased tumor growth and progression. The exogenous administration of IGF-1 partially withdraws all these anti-proliferative effects, thus confirming its role in cancerogenesis. Moreover, IGF-1 and its downstream signaling factors have also been related to inflammation and resistance to BC therapies [12,13]. 

Due to this evidence, in recent years, nutrigenomics and nutrigenetics have been successfully introduced as helpful resources confirming the role of nutrition in cancerogenesis and then in primary and secondary prevention. 

The standard of care in postmenopausal ER+/progesterone receptor positive (PgR)+ BC patients is an aromatase inhibitor (AI) for 5 to 10 years, based on the risk of relapse [14]. Several authors have reported that AI shows less efficacy in obese women. Pfeiler et al., in their prospective trial, highlighted that overweight patients taking an AI had an increased risk of BC recurrence and death, compared to normal weight patients. Moreover, regardless of size, stage of disease and well-known prognostic factors, the benefits of adjuvant endocrine and chemotherapy were significantly less in the obese population [15]. Up to a half of postmenopausal patients taking AI develop musculoskeletal symptoms, referred to as aromatase inhibitor-induced musculoskeletal symptoms (AIMSS), that in 25% lead to therapy discontinuation [16]. The body mass index (BMI) role in AI-induced arthralgia was investigated in literature, exhibiting divergent results. In their paper, Beckwée et al. described that a BMI of 25–30 is associated with a lower risk compared to a BMI < 25 [17]. Moreover, Crew et al., in their cross-sectional survey conducted in 200 patients taking an AI in adjuvant setting, showed that overweight patients (BMI 25 to 30 kg/m^2^) were less likely to complain of joint pain compared with women with a normal BMI or obesity (BMI > 30 kg/m^2^) [18]. Finally, Sestak and colleagues, in their randomized trial, confirmed obesity as a potential risk factor for arthralgia [19].

Beyond the diet, moderate-intensity physical activity (PA) is the most important lifestyle factor in the outcome of BC, able to reduce mortality [20,21]. Current exercise prescription recommends that survivors of cancer engage in at least 150 min of moderate- or 75 min of vigorous-intensity aerobic exercise per week to obtain health benefits. Moreover, the exercise program should also include biweekly resistance training and daily muscle stretching [22,23]. The HOPE study demonstrated that patients who were treated with AI and were following a training program based on 2.5 hours of moderate exercise a week obtained a 3% reduction in body weight after three months compared to those who maintained a sedentary lifestyle [24]. Moreover, a moderate PA concurs with a 40% reduction in BC relapses [25] and a 37% reduction in mortality [26].

The principal aim of the intervention is to promote weight loss through diet and physical activity in overweight and/or high-risk breast cancer survivors. Moreover, implementing a “lifestyle interventions program” in clinical practice can lead to improvement in psychophysical well-being and favor the correction of cardiovascular risk factors, potentially translating into a prognostic advantage.

## 2. Materials and Methods

### 2.1. Study Population

Study participants were recruited among all overweight and/or high-risk BC patients treated at the Breast Cancer Unit Azienda Ospedaliera Universitaria (AOU) Ospedali Riuniti Ancona of Marche (Italy), after breast cancer treatments (surgery, chemotherapy and/or radiation therapy) had been completed. Women undergoing hormone therapy were eligible for the study. Groups of eligible participants were invited at the Department of Medical Oncology AOU Ospedali Riuniti Ancona for further eligibility assessment and enrollment. After an information meeting, they were invited to sign a privacy disclosure. Enrollment began in January 2014 and ended in March 2017.

The study included women aged 18 years or older who had been treated for a histologically confirmed, invasive, non-metastatic breast cancer and had completed their main cancer treatment (surgery, chemotherapy, radiation therapy) for more than 6 months, who had at least one of the following risk factors at diagnosis: BMI ≥ 25 kg/m^2^, high testosterone levels (testosterone ≥ 0.4 ng/mL or 1.152 nmol/L), high serum insulin levels (serum Insulin ≥ 27 uU/mL or 50 pmol/L) or diagnosis of metabolic syndrome (MS) based on National Cholesterol Education Program (NCEP) criteria (Adult Treatment Panel, 2005 guidelines). Only women who were able and motivated to participate in the intervention and who signed the privacy disclosure were enrolled.

Women satisfying any of the following criteria were not eligible for the study: previous severe medical condition(s); advanced age impeding their adherence to the planned study schedules; contraindications to exercise due to any heart condition, stroke, chest pain during activity or rest; severe hypertension and any orthopedic complications that would prevent optimal participation in the physical activities prescribed; unable to walk for exercise (self-reported). 

Patients were not allowed to change medical drugs for anxiety, depression, glycemia, cholesterol and triglycerides unless strictly necessary for medical reasons. 

### 2.2. Study Outcomes

The main aim of the study is to evaluate the impact of a 12-month supervised intervention program on body mass change in overweight and/or high-risk BC survivors. As secondary outcomes, we examined the impact of the intervention on long-term weight control or maintenance, physical activity levels (PALs), dietary intake, health status and quality of life. We also examined the effect of intervention on blood lipid profile (total cholesterol and triglycerides), insulin, glycemia and psychophysical well-being at 6- and 12-month follow-up visits. 

### 2.3. Lifestyle Intervention

The intervention focused on weight loss combining a reduced energy intake, an increase in physical activity level and reduction in sedentary behaviors. At the same time, patients were evaluated by oncologists, dieticians, physiatrists and an exercise specialist. Dieticians’ intervention consisted of an individualized Mediterranean diet (1400–1600 kcal/day) with a low glycemic index and rich in vegetables (processed food excluded). Key strategies emphasized the reduction of energy density by low-energy dense foods (e.g., water- and fiber-rich vegetables and fruits), limiting the intake of foods and beverages high in fat and added sugar and limiting portion sizes. Patients noted the trend in body weight and adherence in a food diary. All participants were monitored to prevent any possible deficiency or inadequate nutrient and/or caloric intake by means of an evaluation of their dietary intake (dietary recalls and/or food diaries). Physiatrists and an exercise specialist prescribed physical activity on an individualized level based on capabilities, lifestyle pattern and preferences to increase physical activity levels and to reduce sedentary time. The general exercise prescription planned were moderate to vigorous intensity (equivalent to 3–9 MET-hours per week), regular frequency (3–5 times/week) and involving aerobic, resistance or mixed exercise types. In accordance with the FITT-VP principles and the physiatrists, the exercise specialist illustrated to each patient the types of exercise that could be performed and discussed with them the intensity of different activities. In order to reach the exercise guidelines goals for cancer survivors, the exercise specialist was available for one morning a week at the gym of the Neurorehabilitation Clinic University Hospital of Ancona, to train patients who wanted to participate voluntarily and for free. In this session, the exercises were explained and demonstrated by the exercise specialist and the women were invited to practice the exercises for a few minutes. Participants familiarized themselves with aerobic, resistance and stretching exercise with or without equipment (e.g., treadmill, medicine ball, elastic band, etc.). On each follow-up, physiatrists and the exercise specialist could decide to modify the PA goals based on the adherence obtained and patient feedback. Participants were instructed to report any problem or adverse events immediately to the clinical staff. Participants were also monitored for injuries or problems associated with increased physical activity level. 

### 2.4. Assessments

Body mass and waist circumferences were measured at baseline (T0) and at 3- (T1), 6- (T2) and 12-month (T3) follow-up visits. Height was measured at baseline. A fasting blood sample was collected at T0, T2 and T3 to measure lipid profile (total cholesterol and triglycerides) and insulin, glycemia and testosterone levels. Body composition was assessed at T0 and T1, T2 and T3 follow-up visits, using bioelectrical impedance vector analysis (BIVA). PAL was assessed using the International Physical Activity Questionnaire-Short Form (IPAQ-SF) [27,28]. The questionnaire referred to the past 7 days and responses were converted to Metabolic Equivalent Task minutes per week (MET-min/wk) [27] according to the IPAQ scoring protocol: total minutes over the past seven days spent on vigorous activity, moderate-intensity activity and walking were multiplied by 8.0, 4.0 and 3.3, respectively, to create MET scores for each activity level. MET scores across the tree sub-components were summed to indicate overall physical activity [27]. In addition to the IPAQ-SF, the interview used the six-minute walk test (6MWT) distance [29] as an objective measure to assess the physical fitness of patients. The IPAQ-SF questionnaire and 6MWT were performed at T0, T2 and T3. Dietary intake was measured at T0, T1, T2 and T3 using the validated and self-administered food frequency questionnaire (FFQ) developed for the European Prospective Investigation into Cancer and Nutrition Italian section (EPIC) study [30,31]. It records daily intake of foods and nutrients over the previous year. It consists of 15 sections (first course, second course, side dish, fruit, etc.) and contains 254 questions investigating a wide range of food items. Psychological status was assessed via the Hospital Anxiety and Depression Scale (HADS) at T0, T1 and T3. This tool consists of two scales with 7 items, one for assessing anxiety and one for assessing depression, with scores from 0 to 3 for each item. Scores less than or equal to 7 are considered normal, 8 to 10 are borderline and scores greater than or equal to 11 are indicative of clinically relevant anxiety or depression, or both. For the purpose of the study, we considered normal patients with scores ≤ 10.

Change in functional status between T0, T2 and T3 was assessed using the physical function subscale of the Medical Outcomes Study Questionnaire Short Form-36 Health Survey (SF-36) questionnaire as an indicator of overall physical function and pain. The physical function subscale assesses the impact of health on the performance of activities ranging from basic self-care to vigorous PA, and has been widely used with good construct validity and sensitivity to change. Table 1 summarizes the required assessments of lifestyle program. 

### 2.5. Statistical Considerations

The present study was designed as a mono-institutional, single-arm educational trial. The study was conducted over a 2-year period, during which patients received a 12-month intervention and was followed until the end of the study period or for a maximum of 10 years after diagnosis of BC. The mono-institutional study design was characterized by uniform diagnostic procedures, standard treatment protocols and central pathological evaluation of surgical specimens, thereby limiting variations in the determination of major clinical and pathological factors that determine disease outcome. It also ensured a uniform follow-up of patients for the study-specific endpoint. Statistical analysis was performed with the MedCalc package (MedCalc^®^v9.4.2.0 Software, Ostend, Belgium). All continuous variables (anthropometric, biomedical, nutritional and psychological measurements) were checked for normality and transformed as needed. The χ^2^ test was used to analyze the differences and correlations between variables. Analysis of variance (ANOVA) for repeated measure was performed to obtain intergroup comparison; *p* < 0.05. Analysis of prognostic factors was not performed due to the low number of events (relapses and/or deaths).

## 3. Results

### 3.1. Baseline Patients Characteristics

From a database of 1320 BC patients referred to the Breast Unit AOU Ospedali Riuniti Ancona of Marche Region (Italy) from January 2014 to March 2017, we identified 160 potentially eligible BC patients (12% of all). Twenty-four patients declined to participate while 38 were ineligible. Finally, 98 female BC patients were included (Figure 1). The median age was 56 years (27–75 years) and 94.9% of participants had a BMI ≥ 25 kg/m^2^. At T0, blood testosterone and insulin levels were higher than the normal in 17.6% and 18.9% of patients, respectively. Moreover, 46.9% of patients had hypercholesterolemia while 13.3% had hypertriglyceridemia and 21.4% of patients had diagnosis of MS. At T0, only 34 subjects (34.7%) practiced regular PA, and only 17.3% of enrolled patients passed the 6MWT. In addition, 79 women (80.6%) had a daily calorie intake higher than 1600 Kcal/day. Total HADS score was abnormal in 46.3% of patients. Some 32.7% of patients referred moderate/strong physical pain (mainly joint pain). In our sample, 79.6% and 20.4% of patients underwent quadrantectomy and mastectomy, respectively. Additionally, 85.7% of BC patients had a luminal phenotype that required adjuvant endocrine therapy (72.3% aromatase inhibitors-based and 27.7% tamoxifen-based) and 62.2% of patients also received adjuvant chemotherapy. After a median follow-up of 42 months, two subjects (2%) developed a disease progression. All data are summarized in Table 2.

### 3.2. Primary Outcome: Body Weight Variation at T1, T2 and T3

At T1, the percentage of overweight BC patients was significantly reduced (patients with BMI ≥ 25 decreased from 94.9% to 81.6%; *p* = 0.0022). At T2, the percentage of patients with BMI ≥ 25 was further reduced (81.6% vs. 68.4%, *p* < 0.0001). At T3, overweight patients were 63.2% (vs. 68.4%, *p* < 0.0001). Similarly, the percentage of patients with waist circumference (WC) ≥ 80 cm at T0 (89.8%) was significantly reduced at T1 (88.8%, *p* = 0.0003), T2 (81.6%%; *p* < 0.0001) and T3 (65.3%; *p* < 0.0001) (Figure 2). 

### 3.3. Secondary Outcomes at T2 and T3

All patients followed diet prescription after T2 and T3 while only some patients reported an increase in PA levels and reduction in physical inactivity from T0 to T2 and T3. An increase in PA levels occurred in 50 and 66 patients after T2 and T3, respectively. However, the percentage of patients who passed 6MWT was significantly higher at T2 (53.1% vs. 17.3%; *p* = 0.0157) and T3 (73.5% vs. 53.1%; *p* < 0.0001). 

At T2 and T3, patients reported a significant reduction in glycemia and insulinemia. In total, 23.5%, 13.3% and 10.2% of patients had hyperglycemia (≥110 mg/dl) at T0, T2 and T3, respectively. Similarly, the percentage of patients with hyperinsulinemia decreased from 20.6% at T0 to 8.8% and 2.9% at T2 and T3, respectively (Figure 3).

Regarding changes in blood lipid profile, cholesterol and triglycerides were not significantly different from T0 to T2 (46.9% vs. 42.9%, *p* = 0.25; 13.3% vs. 14.3%, *p* = 0.76), while from T2 to T3, the difference is statistically significant (35.7% vs. 42.9% of patients with hypercholesterolemia, *p* < 0.0001; 10.2% vs. 14.3% of patients with hypertriglyceridemia; *p* < 0.0001) (Figure 4).

The percentage of patients with abnormal total HADS score significantly decreased from T0 to T2 and T3 (46.3% vs. 34.3% vs. 20.9%; *p* < 0.0001). Patients with abnormal anxiety HADS scores decreased from 25.4% to 20.9% at T2 (*p* = 0.0064) and from 20.9% to 13.4% at T3 (*p* = 0.0064).

The percentage of patients with abnormal depression HADS scores progressively reduced at T2 (12% vs. 7.5%, *p* = 0.19) and T3 (7.5% vs. 4.5%, *p* < 0.0001) (Figure 5). Finally, the percentage of patients without physical pain (mainly joint pain) was significantly higher at T2 (75.5% vs. 67.3% at baseline; *p* = 0.008) and at T3 (82.6% vs. 75.5% at 6 months; *p* < 0.0001) (Figure 6).

### 3.4. Analysis of Variance (ANOVA)

The association between BMI, WC, insulin, testosterone, HADS, pain, glycemia, cholesterol and triglycerides was assessed using factorial analysis of variance (ANOVA). For the ANOVA, the log-transformed BMI was used as a continuous dependent variable and WC, insulin, testosterone, HADS, pain, glycemia, cholesterol and triglycerides as categorical variables. Table 3, Table 4 and Table 5 show ANOVA analyses at T0, T2 and T3. 

## 4. Discussion

Healthy lifestyle can play a crucial role in BC prevention and the modifiable factors that can have a strong impact on the disease, including PA, BMI, blood concentrations of hormones/pro-inflammatory cytokines and diet. An increasing number of studies worldwide highlight the importance of lifestyle in counteracting BC, and there is a large body of scientific evidence showing that lifestyle plays a fundamental role in the promotion of health and well-being, prevention [32,33,34] treatment and reducing the risk of recurrence [35]. 

Based on these emerging data, physicians and other health professionals have a moral, ethical and professional obligation to inform patients of the risk of being physically inactive and to provide them with the right exercise prescription. In fact, although the healthcare community often provides recommendations on lifestyle for patients with chronic-degenerative diseases, it has not yet developed a health prevention plan for cancer patients after surgery. To date, PA is recommended based on general guidelines provided by the American College of Sports Medicine [22] for a specific population and healthcare professionals, assuming that patients fully adhere to the exercise protocols provided to them. However, this does not happen in practice, and only 8.9% of female BC survivors meet the current guidelines on aerobic and resistance exercise [23]. Based on available data, exercise adherence represents a critical point influenced by several factors [36,37]. 

Interestingly, the healthcare professionals can facilitate the ability to schedule exercise, but this is not sufficient if not supported by continuous scientific information and support in the clinical practice. In this study, the PAL increased in 50 and 65 of the patients after 6 and 12 months, respectively. This supports the feasibility of a structured lifestyle program that includes personalized educational intervention and general advice on PA and exercise.

Moreover, an inconsistent irregular exercise program could pose cardiovascular risks and potentially damage patient health rather than providing a powerful means of prevention as demonstrated in the literature [38]. To be successful in the promotion of exercise, it is essential that, alongside the patient’s clinical program and initial counseling session on PA, a structured educational program is provided. Our study demonstrates the importance and effectiveness of a specific exercise education provided in the hospital, prescribed and supervised by a qualified and specialized exercise trainer, who starts working with the patient at the end of her therapeutic regime and then continues to provide support to help the patient gradually move towards an active lifestyle. Numerous studies have highlighted the low PALs in BC patients at different stages of disease [39] in clinical and emergency contexts [40,41,42]. 

In this regard, clinical and epidemiological studies have also identified that adequate PALs have a favorable effect on body weight among BC patients and this is amplified if dietary habits are also modified in addition to changing exercise habits [43].

Our study showed the great association between diet and specific PA advice with the reduction of BMI. Many mechanisms associating adiposity with an increased risk of BC are listed: first of all, the chronic inflammatory state described in obese patients can lead to DNA damage. In addition, adipose cells down-regulate the immune system and promote cancer invasion and metastases by producing inflammatory cytokines and mediators. Moreover, adipose tissue is an endocrine tissue: mature adipocytes in lean adipose organ produce adiponectin, an antimitogenic hormone. Finally, in obese patients, preadipocytes secrete the proangiogenic and pro-mitogenic leptin [30].

Obesity, together with several metabolic abnormalities such type 2 diabetes mellitus, hypertension and dyslipidemia, realizes MS [31], which, according to Esposito’s systematic review and meta-analysis, is associated with a 52% increased risk of BC in postmenopausal women [44]. MS represents a pro-oncogenic environment: patients have a singular insulin resistance, then realizing hyperinsulinemia. Several epidemiologic studies demonstrated that hyperinsulinemia increases morbidity and mortality, representing a cardiovascular risk factor and leaving the patient at higher risk of developing a number of tumors (breast, colorectum, liver and pancreas) [45]. 

In our study, patients revealed a BMI reduction, as well as a waist circumference reduction, as expected. In both parameters, the difference was statistically significant in intervals of 0–3 months, 3–6 months and 6–12 months. The same results were obtained in the DIANA-5 (Diet and Androgens) study, a multicenter controlled trial in which the authors randomized 1208 women to an intensive exercise and diet modification or to a standard group, highlighting that adhesion to MD and healthy lifestyle can reduce BC recurrence, weight and waist circumference [45].

The χ^2^ analysis showed a statistically significant reduction in blood testosterone concentrations (*p* ≤ 0.0001) at all time intervals, and in total cholesterol and triglycerides, which significantly reduced in the period between T2 and T3 (*p* ≤ 0.0001 for both), as shown in literature. Duggan et al. conducted an analysis in nearly 500 post-menopausal obese women, and found that sustained weight reduction turns into a significant reduction in circulating hormones levels. Participants were randomized into three lifestyle-changing intervention groups— diet, combining diet and exercise, and control—and followed them for one year. After 30 months, women who maintained weight loss showed a statistically significantly greater decrease in estradiol and free testosterone levels, along with an increase in Sex Hormone Binding Globulin, compared to women who did not pursue such weight loss [46].

There was a statistically significant difference between the basal blood glucose and insulin levels at T0 and at T2 (*p* = 0.0405 and *p* < 0.0001 respectively); this is in accordance with the literature: Pasanisi et al. conducted a randomized controlled trial evaluating the adhesion to MD and the IGF-1 levels and other markers of insulin resistance in BRCA mutation carriers. In the trial, 203 patients were randomized and those in the experimental arm showed a reduction in IGF-1 levels [47]. 

We assessed whether the lifestyle program improves psychophysical well-being and is associated with a decrease in the total HADS score (*p* < 0.0001) and the anxiety HADS score (*p* = 0.0064), reducing the patient’s need for psychological support. Moreover, from T2 to T3, there are also statistically significant differences for the depression HADS score (*p* ≤ 0.0001). Aydin et al. analyzed the relationship between lifestyle and depression, highlighting that healthy nutrition is linked with lower levels of depression in women who were treated for BC [48]. 

Aromatase inhibitors (AI)-induced arthralgias is more frequent in obese patients. In clinical practice, patients taking AI usually complain of arthralgia, and this can lead to early discontinuation of treatment. Women adopting the lifestyle program revealed an improvement in musculoskeletal symptoms at T2 (*p* = 0.008) and at T3 (*p* ≤ 0.0001), showing that a healthy lifestyle could increase patient compliance and therapy adherence. 

Our study has some limitations, the first being the limited sample size and the heterogeneous population. It would be interesting to evaluate the same results in large and prospective trials. The second limitation is the short intervention duration and follow-up—it could also be useful to update patient survival and relapse in coming years. This is in line with evaluating, in the future, whether the correction of cardiovascular risk factors, weight loss and regular PA contributes to decreasing the incidence of BC recurrence and to reduce mortality. Third, the absence of a control group, the unpredictable motivation and subjective variables (such as arthralgia, anxiety and depression) and the lack of evaluation of other risk factors affecting BC patients and their quality of life (such as quality of sleep, the stability of the sleep/wake rhythm, shift work, nights shifts, etc.). 

## 5. Conclusions

Our study suggests the integration of exercise programs and dietary intervention into the treatment of overweight high-risk BC patients, as healthy lifestyle reduces risk factors involved in BC recurrence and ensures psycho-physical well-being, potentially translating into a prognostic advantage. The positive effects obtained in terms of physical and mental conditioning were sufficient to avoid cases of drop-out during the entire period of intervention. For the BC patients, the “post-surgery” phase is a long and difficult period from a psychological point of view, and it includes frequent checkups and follow-ups. Thus, the inclusion of lifestyle program can help to create a physio-psychological support network during this delicate/difficult phase. A healthy lifestyle promotion that includes specific information on PA and nutrition with periodic follow-ups also provides opportunities to socialize in groups, a fundamental element from a psychological standpoint. To be successful in the promotion of diet and exercise, it is essential that a structured educational program is needed. Our study demonstrates the importance and effectiveness of an exercise and diet program provided in the hospital, prescribed and supervised by qualified specialists, who start working with the patients at the end of their therapeutic regime and then continue to provide support to help them to move gradually towards an active lifestyle. This modeling approach in the promotion of a healthy lifestyle during therapeutic treatment, with suitable space within the clinical facilities where patients can move and train safely, should become an essential part of cancer care. A multidisciplinary approach is strictly needed to allow greater adherence to healthy attitudes in high-risk BC patients.

## Figures and Tables

**Figure 1 cancers-13-05539-f001:**
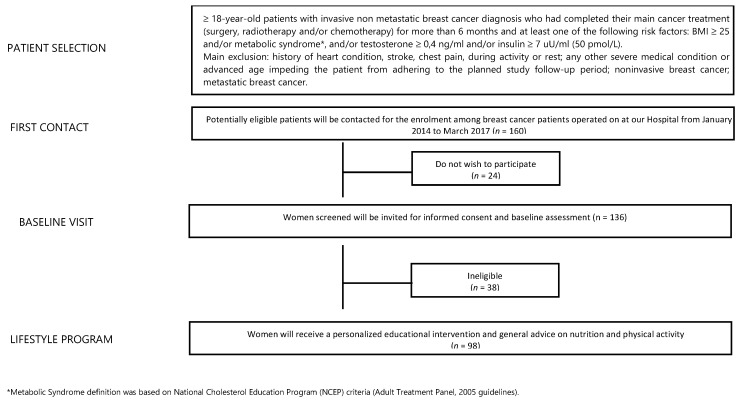
Study flowchart.

**Figure 2 cancers-13-05539-f002:**
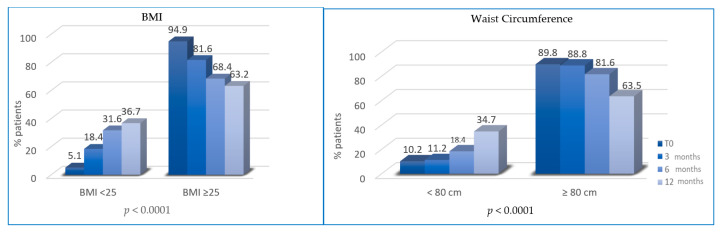
Variation of body mass index (BMI) and waist circumference (WC) between T0, T1, T2 and T3.

**Figure 3 cancers-13-05539-f003:**
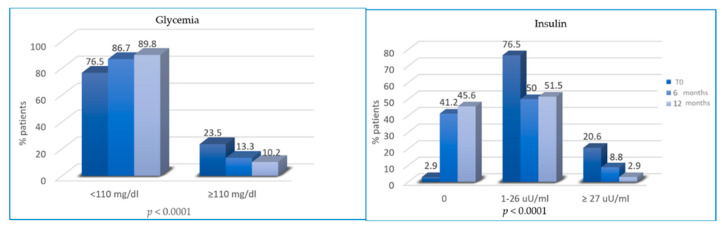
Variation of glycemia and insulin from T0 to T2 and from T2 to T3.

**Figure 4 cancers-13-05539-f004:**
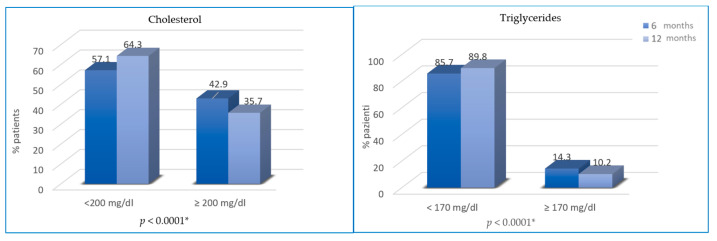
Variation of cholesterol and triglycerides from T2 to T3. * not statistically significant between T0 and T2.

**Figure 5 cancers-13-05539-f005:**
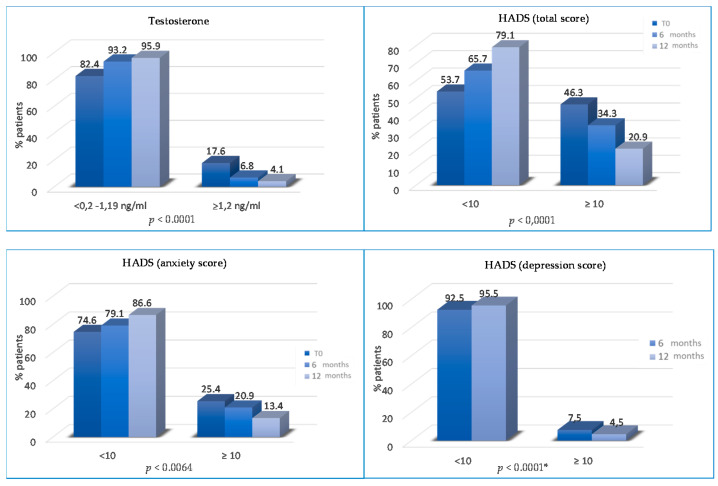
Variations of testosterone and HADS SCORES (total, anxiety and depression) from T0 to T2 and/or from T2 to T3. * not statistically significant between T0 and T2 months.

**Figure 6 cancers-13-05539-f006:**
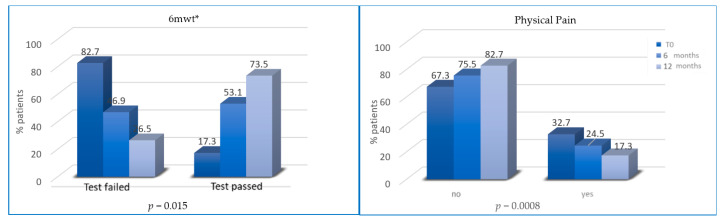
Variation of 6MWT (percentage of tests passed/failed) and physical (joint) pain from T0 to T2 and from T2 to T3. * 6mwt = six minutes walking test.

**Table 1 cancers-13-05539-t001:** Study assessments.

Assessments	Instrument	Visits			
		Baseline	3-Month	6-Month	12-Month
Body composition/ Anthropometrics	Body mass, BMI, waist, hip and limbs circumferences, skinfold thickness	✓	✓	✓	✓
Blood sample collection	Lipid profile (total cholesterol and triglycerides), insulin, glycemia, testosterone	✓		✓	✓
Physical activity level and health-related parameters	IPAQ-SF and 6MWT	✓		✓	✓
Dietary habits	FFQ	✓		✓	✓
Psychological status	HADS	✓		✓	✓
Functional status	SF-36	✓		✓	✓

Note: ✓, timeframes of follow-up measures. Abbreviations: BMI, body mass index; IPAQ-SF, International Physical Activity Questionnaire-Short Form; 6MWT, six-minute walk test; FFQ, food frequency questionnaire; HADS, Hospital Anxiety and Depression Scale; SF-36, Medical Outcomes Study Questionnaire Short Form-36 Health Survey.

**Table 2 cancers-13-05539-t002:** Patients’ clinical characteristics and stratification factors at T0.

Patients Characteristics		Tumor Characteristics		Treatments Characteristics		Clinical Data		Assessments	
	Median (%)		Median (%)		Median (%)		Median (%)		Median (%)
Age (years)		Histological type		Adjuvant radiotherapy		Testosterone * (ng/mL)		Physical pain (SF-36)	
≤65	74 (75.5)	Ductal	83 (84.7)	No	14 (14.3)	≤1.19	61 (82.4)	No/mild	66 (67.3)
≥65	24 (24.5)	Lobular	9 (9.2)	Yes	84 (84.7)	≥1.2	13 (17.6)	Moderate/strong/very strong	32 (32.7)
		Mixed	5 (5.1)						
		Other types	1 (1.0)						
Menopausal status		Grading		Neo-adjuvant chemotherapy		Insulin * (microU/mL)		Total HADS Score **	
Pre/Peri-menopause	27 (27.6)	G1	18 (18.3)	No	96 (98.0)	0	2 (2.7)	<10	36 (53.7)
Post-menopause	71 (72.4)	G2	37 (37.8)	Yes	2 (2.0)	1–26	58 (78.4)	≥10	31 (46.3)
		G3	43 (43.9)			>27	14 (18.9)		
Type of surgery		Tumor phenotype		Adjuvant chemotherapy		Glycemia (mg/dL)		Anxiety HADS score **	
Quadrantectomy	78 (79.6)	Luminal A-like	31 (31.6)	No	37 (37.8)	<110	75 (76.5)	<10	50 (74.6)
Mastectomy	20 (20.4)	Luminal B HER2 negative-like	40 (40.8)	Yes	61 (62.2)	≥110	23 (23.5)	≥10	17 (25.4)
		Luminal B HER2 positive-like	13 (13.3)						
		HER2 enriched-like	5 (5.1)						
		Triple negative	9 (9.2)						
BMI (kg/m^2^)		Staging (Ajcc/UICC 2010) (*n*)		Adjuvant trastuzumab		Triglycerides (mg/dL)		Depression HADS score **	
<25	5 (5.1)	I	47 (48.0)	No	80 (81.7)	<170	85 (86.7)	<10	59 (88.1)
≥25	93 (94.9)	II	36 (36.7)	Yes	18 (18.3)	≥170	13 (13.3)	≥10	8 (11.9)
		III	15 (15.3)						
Waist circumference (cm)				Adjuvant endocrine therapy		Total cholesterol (mg/dL)		Practice of regular physical activity	
<80	10 (10.2)			Tamoxifen	11 (13.2)	<200	52 (53.1)	No	64 (65.3)
≥80	88 (89.8)			AI	54 (66.3)	≥200	46 (46.9)	Yes	34 (34.7)
				AI + LH + RHa	5 (6.0)				
				Tamoxifen + LH-RHa	12 (14.5)				
Metabolic syndrome				Relapse				Usual diet (Kcal/day)	
No	77 (78.6)			No	96 (97.6)			<1600	19 (19.4)
Yes	21 (21.4)			Yes	2 (2.4)			≥1600	79 (80.6)
								Physical function (6MWT)	
								Passed	17 (17.3)
								Failed	81 (82.7)

Note: Values are reported for 98 participants if not differently specified. * Data on 74 participants; ** data on 67 participants; %, percentage of total number of participants. Abbreviations: BMI, body mass index; SF-36, Medical Outcomes Study Questionnaire Short Form-36 Health Survey; HADS, Hospital Anxiety and Depression Scale; 6MWT, six-minute walk test; AI, aromatase inhibitors, LH-RHa, luteinizing hormone-releasing hormone agonist.

**Table 3 cancers-13-05539-t003:** ANOVA analysis of variables at baseline (T0).

Body Mass Index (T0)						
Variables	Levene’s Test *p*-Value	Degree of Freedom	Sum of Squares	Mean Squares	F	ANOVA F-Test *p*-Value
Waist Circumference	0.160	97	4.74	0.07	0.544	0.463
Insulin	0.661	73	3.78	0.06	0.098	0.907
Testosterone	0.001	73	3.78	0.21	3.118	0.082
Hads Score	0.364	66	2.86	0.05	0.206	0.652
Joint Pain	0.214	97	4.74	0.07	0.377	0.541
Glycemia	0.007	97	4.74	0.13	1.609	0.208
Total Cholesterol	<0.001	97	4.74	0.27	4.794	0.031
Triglycerides	0.379	97	4.74	0.06	0.204	0.653

Abbreviation: HADS, Hospital Anxiety and Depression Scale.

**Table 4 cancers-13-05539-t004:** ANOVA analysis of variables at 6 months (T2).

Body Mass Index (T2)						
Variables	Levene’s Test *p*-Value	Degree of Freedom	Sum of Squares	Mean Squares	F	ANOVA F-Test *p*-Value
Waist Circumference	0.509	97	21.19	6.05	36.981	<0.001
Insulin	0.729	67	10.51	0.17	0.080	0.923
Testosterone	0.193	73	13.09	0.33	0.865	0.355
Hads Score	0.500	66	10.48	0.18	0.119	0.731
Joint Pain	0.538	97	21.19	0.24	0.088	0.768
Glycemia	0.094	97	21.19	0.33	0.500	0.481
Total Cholesterol	0.004	97	21.19	0.66	2.082	0.152
Triglycerides	<0.001	97	21.19	0.71	2.279	0.134

Abbreviation: HADS, Hospital Anxiety and Depression Scale.

**Table 5 cancers-13-05539-t005:** ANOVA analysis of variables at 12 months (T3).

Body Mass Index (T3)						
Variables	Levene’s Test *p*-Value	Degree of Freedom	Sum of Squares	Mean Squares	F	ANOVA F-Test *p*-Value
Waist Circumference	0.109	97	22.77	9.62	68.479	<0.001
Insulin	0.021	67	12.23	0.31	0.673	0.514
Testosterone	0.734	73	15.04	0.22	0.037	0.848
Hads Score	0.261	66	11.64	0.24	0.380	0.540
Joint Pain	0.478	97	22.77	0.28	0.171	0.680
Glycemia	0.264	97	22.77	0.29	0.213	0.645
Total Cholesterol	0.001	97	22.77	0.89	2.870	0.093
Triglycerides	0.692	97	22.77	0.25	0.050	0.823

Abbreviation: HADS, Hospital Anxiety and Depression Scale.

## Data Availability

The data that support the findings of this study are available from the corresponding author, M.P., upon reasonable request.

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
