# Peer review of "The Impact of Lifestyle Interventions in High-Risk Early Breast Cancer Patients: A Modeling Approach from a Single Institution Experience"

_cancers, 2021, doi:10.3390/cancers13215539_

Round 1
Reviewer 1 Report
Pistelli et al summarised the need for the promotion of a healthy lifestyle as a primary and secondary prevention tool. They recruited 98 female breast cancer patients between January 2014 and March 2014. They specifically selected cases with at least one risk factor for relapse (BMI>= 25, high testosterone and serum insulin levels, or diagnosis of metabolic syndrome. The patients were enrolled in a lifestyle intervention program including dietary and physical exercise changes. The impact of the 12 months lifestyle intervention was assessed. After 12 months, they report changes in BMI, waist circumference, glucose and insulin levels, Testosterone levels, and Hospital Anxiety and depression scale.
I think that the issues raised in this paper concerning lifestyle interventions in cancer patients are valid and require more attention, but major revisions are needed to improve the quality and readability of this paper.
Major points:
Introductions:
- Some statements were missing citations or unclear. Here are just 2 examples below:
E.g., “Alcohol and animal fats are linked to an increased risk of BC.” Missing citation
- Line 89-90: “The purpose of these disciplines is to analyze the possible interactions between nutrition and the onset, progression and recurrence of tumors”.
This statement is inaccurate as the authors have not looked at the onset of tumors, breast cancer patients were their subjects.
- Line: “In postmenopausal ER+ and/or progesterone receptor (PgR)+ BC patients candidates 92 to receive hormone therapy, the standard of care is an aromatase inhibitor (AI) for 5- 10 93 years, according to risk of relapse [14].”
Unclear. Rewrite needed
Material and methods:
- What was the source of the population? Which hospitals?
- More details about the recruitment and case selection processes needed
- More details are needed about the lifestyle intervention. Which dietary habit questionnaire was used?
- How were blood levels of insulin, glucose, etc assayed?
Statistical analysis:
- The application and presentation of the statistical analysis section need to be rewritten.
- Clear identification of the exposure and outcome variables needed
- Reasons for missing data. Some variables had 2 or 3-time points only.
- The authors’ aim was the investigate the change of the following variables at 4-time points: baseline, 3 months, 6 months, and 12 months.
- BMI
- Waist circumference
- Insulin
- Testosterone
- HADS
- Arthralgia
The correct model to assess 4-time points would be an ANOVA
Conclusion:
It is important for data analysis to be repeated to assess the conclusion section. A major re-write is needed for the methods and results section before the interpretation of Pistelli et al is critically assessed.
Minor comments:
- Spelling and typos
- Abbreviations need to be defined the first time they appear in the main text and not in the general summary
- What is " multidisciplinary ambulatory (Oncology, Dietology and Physiatry)
- A graph showing the change between time points might illustrate the change better than a bar chart
- Improve the readability of table 1:
Suggestions:
1) split into 2 tables ( Table 1: patient and disease characteristics, Table 2: diet and assay results)
Remove the bullet points from the tables
Author Response
Please see the attachment in the box.

Reviewer 2 Report
Authors of the manuscript entitled: “The impact of lifestyle interventions in high risk early breast cancer patients: a modelling approach from a single Institution experience” aimed to evaluate the effects of lifestyle interventions program based on special exercise and nutrition education on weight, psycho-physical wellbeing, blood lipid and hormonal profile among BC patients underwent primary surgery. However at the end of Introduction the Authors state that the primary aim of the study was “to promote a healthy diet and maintenance of the recommended PA level (PAL) in this population”. In my opinion this should not be the aim of the research. Instead the study objectives should be mentioned at the end of Introduction. Although, the subject undertaken by the authors is interesting and worth investigating, the manuscript needs serious improvements before being considered for publication. Specific remarks and comments are listed below:
- Is the change in BMI, WC, glycemia, insulin, testosterone, cholesterol, triglycerides and anxiety and depression related to increased physical activity or is it only caused by modification of the diet? Is there is statistically significant change in those parameters at T2 and T3 in the patients who increased PA? Would be interesting to see the comparison of the improved parameters between subjects who increased PA and did not increase PA.
- More information on PA should be provided for all time points, what was the percentage of PA – yes answers at different time points?
- Did all the subjects follow the diet at different time points? How was it monitored? Is it recorded and does it correlate with the changes in BMI, WC, glycemia, insulin, testosterone, cholesterol, triglycerides and anxiety and depression?
- The statistical analysis of the data raises questions. To compere the difference in variables between time points of the study (dependent samples) other tests should be used instead of χ2 The statement: “The χ2 test was used to compare the difference between the variables.” Is unclear. What groups were compared? What was considered a control group?
- Some of the stratification factors in Table 1 are not described.
- Chapter 3.1.2 named Oncological evaluation contained biochemical and psychological characteristics and not so much oncological evaluation. I suggest changing the name of the chapter.
- Chapter 3.2.1 presents analysis of variables distributed by age only, not by arthralgia, PA and blood tests.
- Page 7, lines 258-260 – the first sentence of the chapter is unclear
- Unify the description of time points. Mark the either 3, 6, 12 months or T1, T2, T3.
- Figure 2 is missing.
- Add the y axis title in figures 3, 4, 5, 6.
- According to table 1 depression HADS score at baseline <10 and >10 had 88% and 12% of subjects, respectively. This is not shown on figure 6.
- Is the format of authors names in the bibliography correct?
- Is the statement that all the 22 authors of the manuscript contributed equally to the work acceptable for the Journal?
Author Response
Please see the attachment in the box

Reviewer 3 Report
Journal: Cancers
Title: The impact of lifestyle interventions in high risk early breast cancer patients: a modelling approach from a single Institution experience
The manuscript of the article is devoted to an urgent topic - the study of the influence of diet and physical activity on the risk of breast cancer recurrence. A number of comments were noted, mainly concerning the design of the study, data processing methods.
- The authors did not take into account a number of factors affecting the risk of breast cancer: the quality of sleep, the stability of the sleep-wake rhythm, shift work, night shifts.
- The authors used indicators of increased risk of breast cancer (testosterone, insulin, triglycerides, cholesterol) as dependent variables, which are only indirect indicators characterizing patients. In order for the authors' data to have a more serious clinical justification, they need to continue the study and present the influence of the studied factors on clinically important indicators (tumor size, presence/absence of metastases, tumor histology, the period from surgery to the appearance of a secondary tumor, a 3-year or 5-year survival rate).
- It is necessary to structure the manuscript more strictly. In the introduction it is necessary to justify the novelty of the study. At the end of the introduction section, formulate a hypothesis and objectives of the study.
- In accordance with the hypothesis to choose an adequate method of statistical data analysis. In modern research, analysis of variance, regression, and logistic regression analyses are most often used, which make it possible to draw a reliable conclusion about the presence/absence of a significant associations between dependent and independent variables, adjusted for concomitant factors.
5. The adequacy of the sample size is questionable. The authors need to provide a statistical justification for the adequacy of the sample size, taking into account the number of variables used in the analysis.
Author Response
Please see the attachment in the box

Round 2
Reviewer 1 Report
The authors have addressed all my comments and suggestions.
However, there are still some spelling, typos, and some sentences that need editing.
For example Table 1:
- No need to add the number of dead patients, as it is only 1 and 12 months is not long enough to follow-up cancer patients
- Make table 1 landscape if all the info kept together with subheadings such patients characteristics, clinical data, treatment
- Metabolic syndrome: YES/NO
Thanks
Author Response
Please see the attachment in the box.

Reviewer 3 Report
Journal: Cancers
Title: The impact of lifestyle interventions in high risk early breast cancer patients: a modelling approach from a single Institution experience
The authors in the corrected manuscript clarified all the unclear questions and answered practically all the comments made. There remains only one comment, to which I have not received a convincing answer: “The adequacy of the sample size is questionable. The authors need to provide a statistical justification for the adequacy of the sample size, taking into account the number of variables used in the analysis."
The small sample size in the manuscript is combined with a very large number of types of analysis and variables used in the analysis, which significantly reduces the reliability of the analysis. Moreover, in the corrected manuscript, the authors added two more types of analysis. I suggest that the authors significantly reduce the number of analyses and variables by removing duplicate analyses and leaving only those that give a clear answer to the objectives of the study.
Author Response
Please see the attachment in the box.
